# Regulating the Revolving Door of Regulators: Legal vs. Ethical Issues

## Elise S. Brezis

Department of Economics, Bar-Ilan University, Ramat Gan 52900, Israel; elise.brezis@biu.ac.il

**Abstract:** This paper analyzes the effects of the revolving door, concentrating not only on the dynamics between regulators and firms but also on whether regulating the revolving door is optimal from the point of view of society. The study explores the trade-off between two interconnected aspects related to the revolving door: the 'lack of competence' and 'greed' of regulators. On the one hand, the revolving door facilitates the recruitment of highly qualified regulators by the government, drawn by the prospect of lucrative future compensation packages. On the other hand, it allows regulators to succumb to greed, enabling them to receive revenues after their term in office. This paper emphasizes that this propensity toward greed can manifest through two distinct channels: 'regulatory capture', which is illegal, and 'abuse of power', which, while legal, is unethical. This paper highlights that distinguishing whether the behavior of the regulator is either unlawful or unethical is of utmost importance for analyzing the optimal policy concerning regulators. On one end, the capture models advocate for regulated oversight of the revolving door to prevent corruption. On the other end, models of abuse of power, characterized by regulators generating 'bureaucratic capital', contribute to the acceptance of the revolving door practice.

**Keywords:** bureaucratic capital; compensation package; corruption; ethics; legal system; revolving door; social norms

**JEL Classification:** H10; H70; O11; O43

## 1. Introduction

This paper examines the question of whether the practice of the revolving door should be subject to regulation. The "revolving door" is a widespread phenomenon in developed countries, in which heads and top officials of state agencies, upon completing their bureaucratic terms, transition into positions within the sectors they previously regulated.

There is a growing interest in understanding how the revolving door impacts the balance of power between regulators and firms, prompting consideration of whether reform is necessary. Zingales (2017, p. 127) who discusses the power of firms writes: "The size of many corporations exceeds the modern state... To fight these risks, several political tools might be put into use, increase in transparency of corporate activities, improvements in corporate democracy, and better rules against revolving doors".

Zingales suggests that the revolving door should be reformed to reduce the power of firms. Should it indeed be reformed? This paper analyzes the effects of the revolving door, concentrating not only on the dynamics between regulators and firms but also on whether regulating the revolving door is optimal from the point of view of society.

To give some examples of revolving doors in the financial sector, Alan Greenspan, Chair of the Fed, moved to the financial firm Paulson and Co; Jacob Lew, US Secretary of the Treasury, went to Lindsay Goldberg, a private equity firm, while Timothy Geithner, US Secretary of the Treasury, went to Warburg, Pincus & Co. Larry Summers, the US Secretary of the Treasury and Robert Zoellick, US Deputy Secretary of State, went to Goldman Sachs. The revolving door phenomenon is also prevalent in many other sectors and also in the EU.[1]

Two primary issues are associated with the behavior of regulators, both of which can be connected to the revolving door. The first issue pertains to the potential for erroneous decisions made by regulators due to their incompetence. Indeed, some regulators appointed to crucial positions lack the qualifications and expertise necessary for effective performance. This challenge is identified as a 'lack of competence'. It may arise when the salary offered for the job is not sufficiently high to attract highly qualified and competent individuals who may prefer employment in the private sector.

This problem can be attenuated by allowing the use of the revolving door. Indeed, the prospect of a high future compensation package, after passing through the revolving door, enables the attraction of competent workers into the civil service.

The second issue arises when some regulatory decisions are not inaccuracies resulting from incompetence but are driven by 'greed'. In this context, regulators take actions aimed at boosting their future compensation, either through the issuance of regulations or through deliberate inaction. Greed may manifest in two forms: 'regulatory capture' or 'abuse of power'.

'Regulatory capture' is when the regulator takes a decision in favor of a specific firm and is then compensated after taking the revolving door and working for the firm that bribed her. That is not legal.

The literature on the capture theory put forward by Stigler (1971), Peltzman (1976), Spiller (1990), and Laffont and Tirole (1996) perceives the power of the regulator as leading to bribery.[2] Indeed, the regulator who is supposed to prevent monopolistic power behaves in her own interest and not in the interest of the people.

This aforementioned behavior is illegal in most countries because it entails collusion between a specific firm and a regulator. So, should we believe that this use of the revolving door is frequent in developed countries? No, because it is somehow peculiar that the regulator will wait until the end of her term in office to be compensated and there are more effective ways for receiving bribes.[3] So, regulatory capture is not a primary motive which could explain the revolving door.

Which motive can then explain the frequent use of the revolving door? This paper emphasizes that the utilization of the revolving door can be explained by greed, manifested as an 'abuse of power' through the creation of 'bureaucratic capital'. In essence, regulators have the potential to increase their lifetime compensation by generating a novel form of capital—'bureaucratic capital'—during their tenure in office.

Bureaucratic capital refers to rules and regulations established by regulators that are deemed unnecessary and have adverse effects on the efficiency of the economy. It also includes her achievement of accumulating an ample phone 'contact list' by investing time in good relationships with the lower-level bureaucracy, ties which will benefit her in the future. As the architect of these rules, regulations, and relationships, the regulator has knowledge of the system, including any loopholes that might exist.[4] This behavior is unethical and should be discouraged. But it is not illegal. The difference between ethical and legal behavior is essential. Public policy should eradicate all unlawful behavior. What about the unethical behavior linked to the bureaucratic capital and the revolving door? Should we reform the revolving door so that regulators may not work in firms they have regulated?

This paper stresses the main difference between unlawful and unethical behavior. Unlawful behavior, as in the case of 'regulatory capture', should not be tolerated even if there are also some benefits from allowing it. In the opposite, unethical behavior, as in the case of 'abuse of power' may be allowed, since it is not illegal. In this case, it is necessary to analyze the trade-off between these two elements linked to the revolving door: 'lack of competence' and 'greed' due to 'abuse of power' by creating bureaucratic capital.

This paper analyzes the trade-off between the positive effects of the revolving door (highly qualified regulators) and the negative effects of the revolving door (bureaucratic capital). We find that the optimal solution is not a corner solution, but to allow the revolving

door practice, and the creation of bureaucratic capital. Therefore, some greedy behavior should be accepted.

In other words, this paper highlights that distinguishing whether a behavior is unlawful or unethical is of utmost importance for analyzing the optimal policy concerning regulators. Models of capture require that the revolving door should be regulated to prevent corruption, while models of abuse of power, in which the regulator creates bureaucratic capital, lead to accepting the revolving door and unethical behavior.

The model is presented in Section 3. The model incorporates the two problems of regulation—competence and greed. The model shows that we should not completely avoid unethical behavior. Section 4 concludes. But why would an economist pay attention to issues of ethics? That is the topic of the next section.

## 2. Ethics and Economics

The literature on economic development, emphasizing the significance of the legal system, has demonstrated that a lack of 'property rights', in particular, and deficiencies in the legal system, in general, impede economic development. In essence, the existence of a reliable legal system is considered a prerequisite for economic progress. However, when it comes to social norms and ethics, a notable distinction arises. While economists universally acknowledge the importance of a robust legal system, ethics has traditionally played a limited role in economic discourse. Although Adam Smith was head of a chair in ethical matters, the primary objective of our economic system is centered on profit, efficiency, and gain, with the economist's curriculum rarely including a focus on ethics or morality. (Smith [1776] 1976a, [1759]1976b).

Furthermore, the literature on economic theory and the normative views of economics tends to advocate that economics should be 'values-free'. In this perspective, values are not considered within the realm of economic solutions. The economic outlook on regulation does not align with ethical inquiries, and ethical norms are absent in models based on the Laffont-Tirole principal–agent model and in the public choice theory of Buchanan (1975, 1987) and Tullock (1965). All agents are homo-economicus.

Even more blunt, Stigler in his *Tanner* lectures on 'Ethics or Economics' wrote: "economists have seldom spent much time exhorting individuals to higher motives or more exemplary conduct" (Stigler 1980, p. 148). Stigler even claimed: the economist "needs no ethical system to criticize error: he is simply a well-trained political arithmetician".[5]

So, when did the Western world begin to raise questions about ethical behavior and social norms in the realm of economics? Ethical considerations gained prominence within the economic discourse when research revealed that immoral behavior detrimentally influences economic growth. Subsequently, the contemporary world started acknowledging the adverse effects of corruption. A pivotal study that significantly influenced policy was Mauro's (1995) paper on corruption. While it might seem commonplace today, the notion was groundbreaking in the 1990s, as corruption had long been regarded as an ostensibly efficient element of society.[6]

In the 1960s, American political scientist Samuel Huntington espoused the economic and social virtues of corruption. He first suggested that corruption provides immediate, specific, and concrete benefits to groups that might otherwise be thoroughly alienated from society. Corruption may thus be functional to the maintenance of a stable political system. (Huntington 1996).

Huntington's second argument in favor of corruption is more subtle. He argued (Huntington 1968, p. 69): "The only thing worse than a society with a rigid, over centralized, dishonest bureaucracy is one with a rigid, over centralized honest bureaucracy." Huntington was not alone in his views about the positive benefits of corruption. Then the wind changed direction. Views on corruption changed, and it became recognized that corruption harms the economy. Today, the number of NGOs addressing the negative effects of corruption are numerous: Transparency International for providing data, the Norwegian NGO U4-Anti-Corruption Center, OpenSecrets in the US, and Corporate-Europe Observa-

tory in the EU. Economic theorists and practitioners of behavioral economics also began to analyze the effects of virtuous behavior on economic success and economic growth.[7] In the next section, we analyze the effect of the revolving door on virtuous behavior. We show that the optimal solution will not be zero tolerance.

### 3. Regulating the Revolving Door: A Model

*3.1. The Structure of the Model*

The "revolving door" is a practice in which heads of state agencies, after completing their bureaucratic terms, enter the very sector they have regulated. The previous literature on the revolving door, linked to regulatory capture, has emphasized corruption motives, which are unlawful in most Western countries.[8] Hence, it is difficult to accept that the majority of the numerous cases of the revolving door can be attributed solely to fraud and unlawful behavior.

A new literature has put the emphasis on the ethical problems of the revolving door. In this literature, conflicts of interest arising from the revolving door are not unlawful, as is in the case of regulatory capture, but still lead to economic distortions, and to an ethical problem. This type of conflict of interest has been coined 'abuse of power' and encompasses the creation of excessive regulation while in public office. As defined by Transparency International (2011), this 'abuse of power' arises when "bureaucrats abuse their power to ingratiate themselves to potential future employers."

The model addresses two issues in regulation previously introduced in the paper: the abuse of power stemming from greed and the lack of competence.[9] First, there are distortions resulting from the abuse of power, typically taking the form of creating bureaucratic capital. Bureaucratic capital is defined as actions and decisions made while in office that enable regulators to capitalize later when joining a firm they had regulated. Examples include cultivating positive relationships with lower-level bureaucracy or developing excessive regulation. These are the negative effects of the revolving door stressed in this paper.

The second regulatory issue pertains to the potential incompetence of regulators. The revolving door can offer a solution to this problem by facilitating the recruitment of competent and qualified bureaucrats. Given the heterogeneity in the abilities of bureaucrats, allowing the revolving door ensures the hiring of high-quality individuals who contribute to more effective regulation, ultimately leading to higher economic growth. These are the positive effects of the revolving door.

In other words, by incorporating positive as well as negative effects of the revolving door, and only those that are not unlawful, this section enables us to analyze the net effects of the conflicts of interest arising from the revolving door on economic growth. This small model enables us to draw two main conclusions.

The first is that it is not optimal to prevent the revolving door, even if it is unethical. The second conclusion is that in equilibrium, the revolving door leads to an excessive level of bureaucratic capital, which in turn leads to over-regulation and to lower economic growth.

In the next sections, we first define what the regulators are doing and then the firms. We then define optimality from the point of view of society.

*3.2. The Demand and Supply of Bureaucratic Capital*

During her time in office, the regulator regulates, but at the same time, she creates *bureaucratic capital*. The bureaucratic capital consists of all the unnecessary regulations she is developing. Bureaucratic capital is supplied by the bureaucrats, while the demand comes from the firms.

Let us start with the regulators, who are appointed by the government to regulate the economy efficiently. Yet the regulators do not merely enact efficient regulation; they also create bureaucratic capital.

Bureaucratic capital therefore enables the regulator to cash in later thereon, after exiting the public sector and joining a firm in the sector she previously regulated.

The regulator maximizes her lifetime income net of efforts, since the production function of bureaucratic capital, H, is a nonlinear function of effort, $E$ (Equation (2)).

The utility function, $V$, she maximizes is:

$$V = \Omega + qH - \lambda E \tag{1}$$

where $\Omega$ is her lifetime salary when appointed as regulator. After leaving her job as regulator, the regulator works in the industry that she regulated. She receives on top of her lifetime salary, a rent, $qH$, which is a function of the amount of "bureaucratic capital", $H$ she has accumulated at price $q$ less her effort, $E$.

The production function of bureaucratic capital, $H$, as a function of effort, $E$, takes the specific form:[10]

$$H(E) = [(1+\gamma)E]^{1/1+\gamma} \quad \gamma > 0 \tag{2}$$

Taking the FOC of Equation (1) with respect to $H$, when $H$ is a function of effort as described in Equation (2), we obtain the optimal level of bureaucratic capital she wants to accumulate:

$$H = (q/\lambda)^{1/\gamma} \tag{3}$$

The second player in this framework are the firms. These firms consist of monopolistic firms each with their own intermediate good, and in consequence, they are regulated by regulators nominated by the government (I follow the model of Romer 1990).

Entrepreneurs maximize profits of the regulated firms. These firms find the knowledge accumulated by the bureaucrat valuable, since the bureaucratic capital under the board's control enables increasing the firm's revenue. In consequence, I develop the demand of these firms for the "knowledge" of the "revolver" regulators, i.e., the bureaucratic capital, $H$.

When a firm, $j$, hires a bureaucrat with a bureaucratic capital, $H_j$, the production of output $j$ becomes more efficient. This is so, because the regulator has knowledge and connections on the system. More specifically, it depends on the relative level of bureaucratic capital by the different regulators of the different firms, since it is only the relative knowledge which matters. So, the production function in sector $j$ takes the form:

$$x_j = k_j \left(\frac{H_j}{\overline{H}}\right)^{\phi} \quad \phi > 0 \tag{4}$$

where $H_j$ is the level of bureaucratic capital produced by the regulator entering firm, j, and as in the Romer model, output is also a function of capital, $k_j$.[11]

So, the profit maximization, $\pi$, for an intermediate good firm, $j$, is:

$$Max\ \pi_j = p_j(x_j)x_j - rk_j - qH_j \tag{5}$$

where the output is $x$, the price of the output is $p$. The two costs of factors of productions are (i) capital, $k_j$, where r is the interest rate, and (ii) the bureaucratic capital, $H$, at cost, $q$. The last term in Equation (5) is the amount paid to the regulator for her bureaucratic capital.

Each firm maximizes profits by finding the optimal amount of output, $x_j$, prices, $p$, and bureaucratic capital, $H_j$.

$$p_j = p = \frac{1}{\alpha}r \tag{6}$$

$$H_j = \overline{H} = \frac{\phi r K}{qA} \tag{7}$$

Equation (7) represents the demand for bureaucratic capital as a decreasing function of $q$. This is the demand from the side of the firms.

Recall that from the side of the bureaucrats, we obtain the supply equation of bureaucratic capital (Equation (3)). By equating demand with supply, we obtain the equilibrium stock of bureaucratic capital, $H^*$:

$$H^* = (\phi rK/\lambda A)^{1/1+\gamma} \text{ and } q^* = [\lambda(\phi rK/A)^{\gamma}]^{1/1+\gamma} \tag{8}$$

Is this level of bureaucratic capital optimal from the standpoint of the economy? In the following section, we show that it is not so, but we also show that the optimal bureaucratic capital level is not zero.

### 3.3. The Effects of Bureaucratic Capital on the Economy

The question to be asked is: Why does a government, for which economic growth is a priority, not find a way to prevent the regulators from creating bureaucratic capital?

We now show that the government indeed does not act to eliminate the revolving door. On the contrary, they find it optimal to allow the regulator to create a certain amount of bureaucratic capital. The solution to ethical problems will never be to choose the extreme. The explanation behind this result is the following: regulators are heterogeneous in their abilities, and more able regulators enable higher productivity and higher economic growth. In order to recruit high-quality regulators, governments must pay them well. An easy way to let regulators have high income is by legislators closing their eyes to the fact that regulators can cash in on the bureaucratic capital they create while serving as heads of agencies.

Thus, a government that wants to maximize the rate of growth of the economy faces a trade-off between, on the one hand, having high-quality regulators, and on the other hand, letting the high-quality regulators create bureaucratic capital, which has negative effects on growth.

Indeed, if the government wants to hire a high-quality bureaucrat, her income must be high. Since wages in the public sector are lower than the reservation wage of the regulator, they must let her accumulate bureaucratic capital so that she will be indifferent between working as a bureaucrat or providing private services as a lawyer or economist. This trade-off is the QH curve in Equation (9), as Figure 1 depicts.

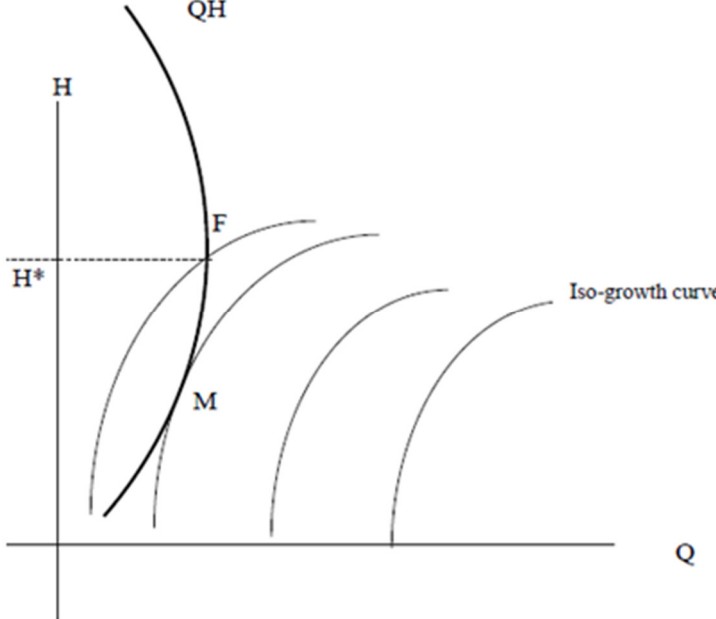

**Figure 1.** The amount of bureaucratic capital in equilibrium, and the trade-off between quality and bureaucratic capital.

This QH curve is the production possibility frontier between bureaucratic capital, *H*, and quality of the regulator, *Q*, in Equation (9). The maximum amount of quality is reached at H = H* (see Appendix A for the details).

$$Q_i = \frac{1}{\xi}[\Omega - \lambda\frac{H_i^{1+\gamma}}{1+\gamma} + qH_i] \text{ (The QH curve)} \tag{9}$$

The QH curve is the curve representing the decision of the regulator.

We are left with the question: Which bundle of quality and bureaucratic capital is optimal for the country? For matter of simplicity, this paper assumes that the goal of the government is to maximize the rate of growth of the economy.

Following Romer (1990), the intermediate goods develop new technology through a sector of R&D, and the final good is produced with labor and the intermediate goods in the following way:

$$Y = L_y^{1-\alpha} \int_0^A x_j^\alpha dj \tag{10}$$

where $Y$ is the output at each period; $L_y$ is the number of workers in the production sector; $x_j$ is the number of intermediate goods from type *j*; and *A* is the level of technology, measured by the range of capital goods available. The power of monopolist firms is also embedded in the parameter $\alpha$.

In consequence, the growth rate in the economy, which is constant (since we focus only on the balanced growth path) is:

$$g = \frac{\dot{A}}{A} = \delta(Q)L_r \tag{11}$$

where $\delta$ is a positive parameter function of the quality of the bureaucrat, $Q$, $\delta' \geq 0$, and $\delta'' < 0$. $L_r$ is the size of the labor force in the R&D sector.

In consequence the rate of growth in the economy is:

$$g = \delta(Q)[1 - \frac{r}{\alpha\delta(Q)} - \frac{q^*H}{\alpha(1-\alpha)\tau}] \tag{12}$$

Equation (12) describes the rate of growth of the market economy as a function of the behavior of the bureaucrat described by *H* and *Q*. The rate of growth in the economy is a positive function of the ability of the bureaucrat, *Q*, and a negative function of the level of bureaucratic capital, *H*. In Figure 1, we present the iso-growth curve as a function of *Q* and *H* (see Appendix B for details). We find that the rate of substitution is positive and the iso-growth curves are concave.

So, the two main equations of this model are the QH function (Equation (9), which describes the production possibility frontier of the regulator faced by the government in terms of quality and bureaucratic capital. The second one is the equation of the iso-growth curves, Equation (12), which describes for each *Q* and *H* chosen by the regulator what the level of growth rate is, *g*, obtained in the economy.

We now determine the amount of bureaucratic capital which leads to the highest rate of growth.

### 3.4. Equilibrium and Optimal Choice

Let us focus on the iso-growth curves in Figure 1. As we move further to the right, an iso-growth curve depicts a higher growth rate (higher quality of a bureaucrat leads to higher growth). In consequence, the highest rate of growth which is also on the production possibility frontier is point M. This is the optimal solution from the point of view of the country. At this point, the level of H is positive and non-zero.

In other words, the optimal level of bureaucratic capital is non-zero. It is in the public interest to allow the bureaucrats to create bureaucratic capital. The solution to an ethical problem is not zero-tolerance.

This result stresses that despite the negative effects of bureaucratic capital on the economy, the economy has an optimal mix of the level of redundant bureaucracy and quality of the regulator. While the government could restrict the possibility of the revolving door, this would mean reducing the quality level of the regulators in the economy, which is not a good solution. This result states that the highest rate of growth is attained when there is creation of bureaucratic capital which is non-zero.

Although the market economy reaches its highest economic growth at point M, we now show that the regulators choose a level of bureaucratic capital which is higher than the one the economy would prefer. The regulator chooses to create bureaucratic capital at the level of H*, which gives her the highest utility. Therefore, in a market equilibrium, the regulators choose the amount of H* and the economy will be at point F. Comparing F to M, we obtain that at F, the amount of bureaucratic capital is higher and the rate of growth is lower.

This fact stresses that the amount of bureaucratic capital chosen by regulators and firms, H*, is higher than that favored by the government and the public. The equilibrium is at a point wherein ability is at its maximum. The public would rather have less bureaucratic capital, even at the price of having less able bureaucrats. The reason we are not at the Pareto optimum of point M in Figure 1 is due to the social waste which bureaucratic capital leads to.

However, the main message of this paper is that at the optimal growth rate (point M), the amount of bureaucratic capital is not zero.

## 4. Conclusions

This paper has analyzed the question of whether the revolving door should be regulated. While the succinct answer is no; the more nuanced answer is intricate. In the past, the literature on the revolving door has predominantly emphasized the regulatory capture model which encompasses corruption, fraud, and unlawful behavior. In those cases, reforming the revolving door becomes imperative and regulatory measures are deemed necessary.

However, this is not the scenario prevalent today. In contemporary times, regulators engage in actions that, while unethical, do not breach legal boundaries. Specifically, regulators take actions during their tenure that enable them to benefit later when joining a firm they once regulated. While the nature of these actions may vary, they all involve what is termed the creation of bureaucratic capital. Consequently, regulators can exploit their former positions to enhance their compensation packages in a perfectly legal yet ethically questionable manner. It is in the government's interest to allow this behavior to attract high-quality regulators who may not find the prevailing salary in the civil service sufficient. The revolving door provides individuals with the opportunity to augment their lifetime compensation, thereby enabling the attraction of higher-quality civil servants.

Consequently, the optimal solution does not lie in adopting a 'zero tolerance' approach towards the ethical concerns associated with the revolving door. The primary conclusion drawn from this paper is that the concise response to the question of whether to regulate the revolving door is "no".

The second conclusion of our paper is related to a taxonomy of social behaviors. Individual actions can be divided into two types: lawful and ethical. The optimal solution to behavior related to legal matters is 'zero tolerance' for fraud, and we should not take into consideration possible trade-offs.

However, policies addressing ethical concerns should refrain from adopting absolute solutions. Ethical dilemmas cannot be effectively resolved through a 'zero tolerance' approach. This principle may apply to many aspects of human behavior; this paper has shown that it is also the case for the revolving door.

Indeed, the rationale for this non-zero solution concerning the revolving door lies in the fact that while bureaucratic capital may be inefficient for society, it enables regulators to be more competent and effective. Imposing strict restrictions on the revolving door would result in a decline in the quality of civil servants, subsequently leading to lower economic growth. There exists a trade-off between competence and the potential for greed.

In the realm of ethical quandaries, extreme solutions are not the remedy. We should allow some bureaucratic capital, as we allow some unethical behavior. The economy needs competent and qualified regulators who can face the challenges of an ever-changing world.

**Funding:** This research received no external funding.

**Informed Consent Statement:** Not applicable.

**Data Availability Statement:** Data sharing is not applicable to this article.

**Acknowledgments:** I wish to thank Jean-Pascal Benassy, Joel Cariolle, Steve Cicala, John de Figueiredo, Catherine Mann, Abraham Ravid, Adam Przeworski, John Van Reenen, and Clifford Winston, as well as the seminar participants at the PSE, the European Public Choice meetings, the NBER conference on Economic Analysis of Regulation, and the U4 workshop for their helpful comments. I thank the Editor and the referees for their remarks and suggestions.

**Conflicts of Interest:** The author declares no conflict of interest.

### Appendix A

One instrument in the hand of government is to appoint regulators who regulate monopolistic firms. Regulators are chosen from among the individuals with specific abilities and skills. These individuals are heterogeneous in their abilities, and as emphasized by Weiss (1980), when ability affects the productivity of a person, then wages are not equal for all: "workers' wage is an increasing function of his ability." Individuals with high ability and quality earn more than ones with less ability.

In consequence, without loss of generality, we assume the following form:

$$W_s = \xi Q_s \tag{A1}$$

where $W_s$ is the income of a specific individual and $Q_s$ is the ability of this individual.

Since quality of the regulator affects economic growth (Equation (11)), the government will choose a regulator with the highest ability possible and who receives an income related to quality given by Equation (A1). We assume that the legislator possesses perfect knowledge of each candidate's ability. In consequence, he knows that the reservation income of the potential regulator is given by (A1) and therefore, the choice faced by the legislator is to hire a regulator with ability such that:

$$Q_i = Max\langle Q_s | \xi Q_s \le V_s \rangle \tag{A2}$$

and the solution is:

$$Q_i = \frac{1}{\xi} V_i \tag{A3}$$

where $V_i$ is the lifetime income of the bureaucrat, $i$. Substituting Equation (1) into Equation (A3), we obtain the relationship between ability and level of bureaucratic capital faced by the political elite(see Equation (9)).

### Appendix B

Based on Romer, the equilibrium is obtained by equating wages earned by workers in the output and the R&D sectors. So, we have:

$$w_r = w_y \tag{A4}$$

where $w_r$ and $w_y$ are wages in the R&D and production sectors, respectively. The total labor force working in the production and the research sectors is constant and denoted by $\overline{L}$:

$$L_r + L_y = \overline{L} \tag{A5}$$

where $L_r$ is the size of the labor force in the R&D sector and $L_y$ is the labor force in the output sector. Since the salary earned by workers in the R&D sector is the value of the patent of their invention, we have:

$$w_r = \frac{\dot{A}}{L_r}P_r \tag{A6}$$

where $P_r$ is the price of a new-design patent and $\dot{A}$ is the number of new inventions developed. Moreover:

$$w_y = (1 - \alpha)\frac{Y}{L_y} \tag{A7}$$

$$rP_r = \pi + \dot{P_r} \tag{A8}$$

where $\dot{P_r}$ is the change in the price of patents and $\pi$ are profits. Since there is no increase in population, along the balanced growth path, output $Y$, and inventions, $A$ grows at the same rate, so that patent prices also are constant, and we obtain:

$$P_r = \frac{\pi}{r} \tag{A9}$$

Moreover, the profit for each of the firms is:

$$\pi = \alpha(1 - \alpha)\frac{Y}{A} - qH \tag{A10}$$

In consequence, we obtain:

$$\frac{(1 - \alpha)Y}{L_y} = \frac{\delta A}{r}[\frac{\alpha(1 - \alpha)Y}{A} - q * H] \tag{A11}$$

Then we obtain:[12]

$$L_y = \frac{r}{\alpha\delta(Q)} + \frac{q^*H}{\alpha(1 - \alpha)\tau} \tag{A12}$$

In consequence, the rate of growth in the economy is Equation (12).

## Notes

[1]   For an empirical analysis of the revolving door in the US financial sector, see Lucca et al. (2014), Shive and Forster (2017), and Brezis and Cariolle (2019); and for the EU, see OECD (2009) and Chalmers et al. (2022).

[2]   The 'regulatory capture' is illegal because the regulator has a special connection with a specific firm. However, some studies show that despite the negative effect of illegality, there may be positive aspects of the revolving door that should not be overlooked (see Che 1995; Agrell and Gautier 2012, 2017; Luechinger and Moser 2014).

[3]   Moreover, the strongest power that firms have over a regulator is not bribes but threats of damaging her reputation and raising doubts about her competence. See also Tabakovic and Wollmann (2018), Yeoh (2019), and Cornell et al. (2020).

[4]   The current literature presents two motivations for the revolving door: "quid pro quo", which is unethical and "regulatory schooling", which encompasses together positive motivations as effective regulation, but also negative ones, as unnecessary and unethical regulation (see Lucca et al. 2014). My paper establishes a clear dichotomy between positive motivations vs. negative, unethical ones. Specifically, it categorizes all forms of unethical behavior under a unified term: 'bureaucratic capital'.

[5]   This differs from Sen's theory that claims that ethical behavior is necessary for the economy: "modern economics has been substantially impoverished by the distance that has grown between economics and ethics" (Sen 1987, p. 7). See also Becker (1976).

[6]   On the effects of corruption, see also Shleifer and Vishny (1993); and related to talent allocation, see Murphy et al. (1991).

7    Lately, game theorists are analyzing the success of societies wherein altruism is part of the system versus societies where it is not, but the goal is not the value per se of being moral and ethical, but rather the fact that when all people are altruistic, society benefits.

8    Note that the wrongdoing arises while the worker is employed as a regulator and it might be difficult to prove it in court. See Kaufmann and Vicente (2011).

9    The model is based on Romer (1990) and Brezis (2017).

10   In other words, $\Omega$ is the lifetime compensation of the regulator, which includes also what she receives after retiring from being a bureaucrat. $qH$ is the 'bonus' she receives from the company which will hire her after leaving office for her over-regulation. This specific equation is based on my previous work in Brezis and Weiss (1997), Brezis and Cariolle (2017, 2019), and on the philosophy about elites presented in Brezis and Temin (2008), Arrow (1994), Hausman and McPherson (1996), Tirole (1986) and also Rajan and Zingales (1998, 2004).

11   Note that if $H_j = \overline{H}$, then the output is just: $x_j = k_j$, no matter the average level of bureaucratic capital. Although having hired a bureaucrat to increase the productivity of the firm may bring advantage from an individual point of view, it is pure waste from a social point of view.

12   Interest rate is determined by the demand of goods, as in Romer (1990, p. 88), and is not a function of the endogenous variables of Equation (9).

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
