# Peer review of "Regulating the Revolving Door of Regulators: Legal vs. Ethical Issues"

_economies, doi:10.3390/economies12010005_

Round 1
Reviewer 1 Report
Comments and Suggestions for Authors
Great article. Set up well and the argument develops logically.
Only comment--the authors note that government salaries tend to be lower on average than the private sector. That seems to be true. I think the authors could expand on this to make their case even stronger.
First, I suspect not only is the average higher in the private sector, there is also great variance. Government workers tend to be on lockstep compensation, regardless of ability. So for very high-competence individuals the opportunity cost of working in the government is very high. So this would amplify the need to allow for bureaucratic capital to attract the most highly qualified individuals.
Second, there is likely a self-selection mechanism at work in sort of a lemons, adverse selection-type model. Those with the greatest competence will tend to leave, but those who are below-average competency will tend to stay in the government because the private sector isn't an attractive alternative.
These are suggestions that I think are consistent with the author's analysis.
Comments on the Quality of English LanguageIt is fine. Some of the sentences seemed to be a bit cluky but I could read and comprehend it fine.
Author Response
I thank him/her for the comments and suggestions. His/her remark on self-selection mechanism is interesting, and can explain the "tilting" of salaries in the public sector. I will think about it. But this is beyond the purpose of this paper. (Tilting is the slope of the increase of salary over time, "the lockstep compensation" as described by the referee).
I could write a footnote on this matter in this paper, but the main conclusion of the paper will be lost, and I prefer to keep the idea for further research. Thanks again for this interesting remark.
Reviewer 2 Report
Comments and Suggestions for Authors
The author should highlight, in the abstract or in the introduction, that the researched problem concerns the legal system and regulations in the United States of America.
The methodology and methods used should be described in detail.
The literature review might be expanded with more contemporary studies.
The conclusions are boldly formulated, which will provoke further discussion.
Additional information or correct citation should be provided, for example, in lines 134-135.
Author Response
I thank him/her for the comments and suggestions. The revolving door is very common in most countries of the Western world. It is called pantouflage in France and Amakudari in Japan.
I wrote in p.1: This revolving door phenomenon is also prevalent in the EU. see Chalmers et al 2022
The methodology is described in section 3, entitled the model.
I thank the referee for suggesting adding some more papers in the review of the literature. I did it. I also have added footnote # 5 quoting the relevant papers.
Reviewer 3 Report
Comments and Suggestions for Authors
The paper proposes an interesting study and attempts to address the ethical and legal problems of the revolving door phenomenon. Although the author attempts, in the introduction, to present a framework of the different aspects of revolving door practices, he/she does so in a rather confused manner, failing to clearly separate the legal from the ethical problems. Regulatory capture is presented as unlawful and thus seems to be equated, in essence, with regulator's corruption. However, there are several forms of regulatory capture that are not unlawful (cf., e.g., Agrell and Gauthier, 2012; Che, 1995; Tabakovic and Wollman, 2018; Yeoh, 2019) and could instead be framed within the discussion around ethics. At the same time, regulatory capture can indeed occur through revolving doors.
Then, as far as ethical issues are concerned, the presentation offered in Section 2 appears in one respect too generic and poorly related to the theme of the paper, namely that of revolving doors; in another, the literature on economics and morality that is considered by the author is very limited (a handful of papers) and referred to old contributions.
Much of the work refers to bureaucratic capital, but it is not made clear whether it said concept is new or not, what is the authors' contribution, etc. (cf., e.g., Che, 1995). On the other hand, the analysis does not specify whether, e.g., the principal-agent model is relevant, whether the revolving door problem concerns (only) independent regulators (i.e., independent regulatory authorities, or (government) regulatory agencies, etc.)
The aforementioned weaknesses of the framework should be clarified in order to appreciate the contribution offered by the author, especially the model proposed and the relevance of the conclusions.
Comments on the Quality of English LanguageEnglish is ok, but this study needs a clearer presentation and the text needs checking.
Author Response
Answers to referee 3
I thank him/her for comments and suggestions. First, the papers cited by the referee were added to the paper.
On the question: can capture be legal? The answer is no. Whenever there is collusion between one firm and the regulator, there is illegal capture, whatever the benefits that can result. Collusion between a firm and a regulator is illegal. Even if there are positive effects. The view that illegal actions should be accepted because it has some positive effects is very problematic.
How did I succeed to produce a model where the firm is gaining from the revolving door without collusion?
Because of the monopolistic competition model. This is the main idea underlying the model. This model will permit to scholars wishing to discuss legal behavior without collusion to do it.
The bureaucratic capital is very different than the human capital suggested by Che.
I wrote: "Bureaucratic capital are all the rules and regulations, enacted somewhat unnecessarily by the regulator.
For theorists, this paper has a new flavor because of the monopolistic competition model. For macroeconomists, the main value added of the paper is to understand the difference between legal and ethical behavior.
Related to the question about "principal, agent and firm". Yes, indeed, but I assume that the principal wants economic growth.
This paper analyzed whether unethical (but legal) actions related to the revolving door should be tolerated. The answer is yes, but not too much. This is the important value added of this paper to the literature on the revolving door.
Round 2
Reviewer 3 Report
Comments and Suggestions for Authors
In this new version, the author revised the abstract and especially the introduction of the manuscript. In this way, the content and scope of the work presented are much clearer than in the previous version. I appreciated this effort.
However, Part II looks the same as before, and of course in my opinion the weaknesses highlighted in the first round (submission) persist with regard to that Part of this work. In addition, there are still some specific aspects that deserve attention and clarification. Among them:
- the abstract attempts to illustrate the revolving door theory, but it could better clarify the contribution of the paper, which, in particular, seems to be in the area of models of "abuse of power" (unethical behavior) rather than “regulatory capture” (unlawful behavior). This might be clarified, for instance, especially revising the text in lines 20-22;
- lines 56-64: I suggest to join these lines in one paragraph, without starting a new paragraph with “We denote…”; also, in line 56: "can be related to the revolving door" instead of "are related…" would seem better;
- lines 85-86 touch on the aspect of "greed": the author should clarify to what extent the element of “greed” is a novelty in the proposed analysis;
- the concept of “bureaucratic capital” plays a crucial role in this work: is this a new concept introduced by the author?
- footnote no. 4 refers to "current literature", generally -- please state references and provide clarification to the points raised above about “greed” and “bureaucratic capital”;
- line 91: perhaps it should be “unnecessary” instead of “unnecessarily” - please check;
- line 104: the author writes “unethical behavior ... allowed" -- but perhaps “unethical behavior ... tolerated" might sound better;
- line 172: the title "Introduction" should be changed with something more specific about the content of this sub-section of the paper;
- lines 177-178: these are unclear; especially the meaning of "… are explained by…" is unclear;
- lines 185-192: should be double-checked to improve exposition; in particular, "Bureaucratic capital also takes the form of developing excessive regulation" seems to be a repetition of what has already been said in lines 185-189;
- line 352: please check font size;
- lines 360-363: the text "The second conclusion of our paper is related to a taxonomy of social behaviors. Individual actions can be divided into two types: lawful and ethical. Our second conclusion is that the optimal solution to behavior related to legal matters is: 'zero tolerance'. Matters of law should be absolutely regulated, and we should not take into consideration possible tradeoffs" does not seem to clearly reflect what is presented in the paper (and it seems to me that, if something is illegal, it is obvious that it is already regulated, by the law, and regulators are no exception).
Suggestion: “The second conclusion of our paper is related to a taxonomy of social behaviors. Individual actions can be divided into two types: lawful and ethical. The optimal solution to behavior related to legal matters is: 'zero tolerance', and we should not take into consideration possible tradeoffs. However, policies related to ethical matters should not try to adopt absolute solutions. Ethical dilemmas …”
lines 399-400: Chalmers et al 2022 looks incomplete.
Comments on the Quality of English LanguageI think minor editing of English language is required. Use of English is very good.
Author Response
Dear Reviewer
I have taken into consideration all your remarks. Please check the attachment.
I thank you for all your suggestions.
